Subject Areas:
mechanical engineering/oceanography/statistics

Keywords:
measurement uncertainty, data decomposition, approximate Bayesian computation, soil moisture, ENSO, strain analysis

Author for correspondence:
A. Alexiadis
e-mail: a.alexiadis@liverpool.ac.uk

# Transformation of measurement uncertainties into low-dimensional feature vector space

A. Alexiadis[1], S. Ferson[2] and E. A. Patterson[1]

[1]School of Engineering, University of Liverpool, The Quadrangle, Brownlow Hill, Liverpool L69 3GH, UK
[2]Institute for Risk and Uncertainty, University of Liverpool, Peach Street, Chadwick Building, Liverpool L69 7ZF, UK

AA, 0000-0002-7285-8896; SF, 0000-0002-2613-0650;
EAP, 0000-0003-4397-2160

Advances in technology allow the acquisition of data with high spatial and temporal resolution. These datasets are usually accompanied by estimates of the measurement uncertainty, which may be spatially or temporally varying and should be taken into consideration when making decisions based on the data. At the same time, various transformations are commonly implemented to reduce the dimensionality of the datasets for postprocessing or to extract significant features. However, the corresponding uncertainty is not usually represented in the low-dimensional or feature vector space. A method is proposed that maps the measurement uncertainty into the equivalent low-dimensional space with the aid of approximate Bayesian computation, resulting in a distribution that can be used to make statistical inferences. The method involves no assumptions about the probability distribution of the measurement error and is independent of the feature extraction process as demonstrated in three examples. In the first two examples, Chebyshev polynomials were used to analyse structural displacements and soil moisture measurements; while in the third, principal component analysis was used to decompose the global ocean temperature data. The uses of the method range from supporting decision-making in model validation or confirmation, model updating or calibration and tracking changes in condition, such as the characterization of the El Niño Southern Oscillation.

# 1. Introduction

Modern measuring equipment allows scientists and engineers to interrogate physical phenomena and behaviours that were

previously unobservable with unprecedented levels of detail. In examples, extending from hydro-ecological observations [1], where aerial and terrestrial measurements are combined to establish a sensing network designed to improve modelling and decision-making, to *in vivo* deformation measurements of the human heart [2] and multiscale measurements of ceramic matrix composites for model validation [3], spatial data are ubiquitous and usually both information rich and infected with uncertainty. Although substantial progress has been made in the efficient handling of large quantities of information-rich data by reducing dimensionality, an unresolved issue is the representation of uncertainty in the corresponding low-dimensional form. This issue is addressed in this article.

When considering information-rich spatial data, like those mentioned earlier, the data handling and analysis can be difficult. Instead of working with a single measurement or a series of measurements, the practitioner is faced with matrices of measurements that are usually defined over a grid with a density, orientation and reference position, which may be constant or varying across the field of the measurements. When temporally consecutive measurements are located on the same or different grids, or when measurements and predictions lie on different grids, it can become difficult to make comparisons between datasets, for example, to validate simulations using measurements from experiments or to identify critical events from evolution of a measurand, such as the strain field during crack initiation and propagation.

One powerful way to address these issues is to employ decomposition techniques to reduce the dimensionality of the data by employing a vector to represent a matrix of data without any of loss of important information [4]. For instance, Karhunen-Loéve decomposition, also known as principal component analysis (PCA), is among a family of such techniques commonly used to describe stochastic processes and random vectors in vibration analysis [5]. In fluid mechanics, Lumley and his co-workers [6,7] used proper orthogonal decomposition to characterize the coherent structure of turbulence. More recently, weighted proper orthogonal decomposition has been used to generate reduced-order models of swirling flow from a turbine [8]; and dynamic mode decomposition [9,10] has been applied to turbulent flows in cavities [11,12]. In structural mechanics, Mottershead and co-workers [13] have pioneered the use of strain decomposition using Chebyshev and Zernike polynomials [14] to decompose strain fields by treating them as images, which elegantly avoids the difficulty of data existing in arrays with different densities and orientations. The vector or series of coefficients resulting from such a decomposition process are often known as a feature vector or shape descriptors. An overview of the use, in engineering, of feature extraction techniques with data fields can be found in [15]. These uses range from tracking damage in composites [16] to model validation [17], known as model verification in meteorological modelling and model updating [13].

Because no measurement is exact, the uncertainty in measurements can influence decisions about the reliability of simulations, the safety of processes and the quality of manufactured components or affect policy making. This article describes a novel method to achieve the transformation of measurement uncertainty into the low-dimensional space or feature vector space without making any assumptions about the probability distribution of the measurement error. The overall process can be viewed as a function whose inputs are the spatial field of measurements, their uncertainty and the decomposition to the low-dimensional space; while the output is a distribution representing the measurement uncertainty in the low-dimensional space. The distribution is obtained via a chain of comparisons of the spatial measurement with synthetically generated datasets, as will be described in the following section. In the subsequent section, the new method is applied to three examples of increasing complexity. The first example consists of fields of displacements in an aluminium beam subject to three-point bending, measured using a digital image correlation system [18] with a measurement uncertainty that was spatially constant. This relatively straightforward example allows an in-depth explanation of the method and a graphical representation of the results using a simple displacement field and then, using a more complicated displacement field, a comparison with previously established recommendations for the validation of computational solid mechanics models. The second example is more complicated with spatially varying measurement uncertainties associated with soil moisture measurements resulting from a Kriging analysis of sparse measurement stations at the Heihe River Basin in China [19]. The final example introduces two additional factors in the uncertainty: gaps in the data and a progressive reduction in uncertainty over time as the measurement acquisition technology is improved. It involves global oceanographic temperature fields obtained monthly over 11 years from 2002 to 2012 [20,21].

Procedures to accurately characterize the uncertainty of a measurement are well established and enshrined in standards, such as the International Organization for Standardization (ISO) 17025 [22], which specify that the calibration of measuring devices should be achieved through a traceable,

continuous chain of comparisons to a primary standard. In engineering, detailed calibration procedures have been developed, for instance, for optical instruments for deformation measurements [23]. Those uncertainties that can be defined *a priori*, for instance, by calibration, are known as type B, while type A uncertainties are the random component of measurements and can be defined based on a series of measurements or repeated observations [24]. Both types of uncertainty can be modelled using probability distributions. When the errors are small and random, they can be represented using the Gaussian or normal distribution [25], which is symmetric and has finite high order moments. A systematic error or bias in the measurement represents a constant offset in the measured quantity relative to the true value, and a correction should be applied before representing it probabilistically. For the case of a Gaussian distribution, the expected value of the measurement is associated with its mean, $\mu$, an estimate of which can be calculated through the arithmetic mean of $n$ observations, while the randomness in the measured quantity is related to the standard deviation, $\sigma$, of the Gaussian distribution. This means that when the two values, the mean ($\mu$) and the standard deviation ($\sigma$) of the Gaussian distribution, are known as the result of a calibration process, then one can estimate the true value of a measurement with a certain level of confidence. In simple terms, the mean, $\mu$, defines the peak value in the bell-shaped curve of the Gaussian distribution, while the random error or standard deviation, $\sigma$, characterizes the width of the curve; to inform decision-making, a 95% confidence interval can be defined as [measured value $\pm 2\sigma$]. In those cases where no knowledge about the probabilistic form of the uncertainty exists, the error can be represented using interval analysis [26], in which the confidence interval is replaced with a range that represents the associated uncertainty, i.e. [measured value $\pm 2u_{\mathrm{meas}}$], where $u_{\mathrm{meas}}$ represents the measurement error, usually obtained from a calibration. This latter approach is recommended in the European Committee for Standardization (CEN) guide for the validation of computational solid mechanics models [27]. Regardless of whether a probabilistic knowledge of the measurement uncertainty is available, it is important to appropriately transform the confidence interval from the measurement space to the feature vector space, i.e. to map the interval into the low-dimensional form of the measurements, to inform reliable decision-making using this data, for instance, about whether to accept the predictions from a computational model. In practice, it is not unusual to have limited or no probabilistic knowledge of the measurement uncertainty, and hence, in the next section, a methodology to achieve this transformation is proposed that makes no assumptions about the probability distribution of the measurement error. Three examples of its application are described in the following section.

# 2. Proposed methodology

Measurements and models are frequently employed to aid decision-making, leading to rational decisions with consequences. This process will involve comparing fields of measurement data with either other sets of measurement data or predictions from a model; and the decision will be influenced by whether the difference between the data fields is significant, which requires knowledge of the associated uncertainty in the data. When the comparison is performed by decomposing the data into a low-dimensional form, then it is necessary to make a quantitative assessment of the difference between the corresponding components in the low-dimensional space, which requires transforming the measurement error into the same space. However, in practice, this transformation of the uncertainty is not performed owing to a gap in knowledge about an appropriate methodology. A new methodology that uses approximate Bayesian computation [28] and results in a distribution representing the measurement and its uncertainty in the component or feature vector space is described later.

The overall methodology for transforming the spatial data and its uncertainty to its low-dimensional form involves the steps shown in figure 1 and the approximate Bayesian computation shown in figure 2. Initially, in figure 1, the dataset is decomposed to represent the data field in a lower dimensional form as a feature vector or set of components. The methodology is independent of the mathematical transformation or decomposition used in this initial stage, and this is illustrated by employing orthogonal decompositions based on Chebyshev polynomials [29] in two of the examples and on PCA [30] in the third example. In figure 2, during the approximate Bayesian computation, the measurement uncertainty in the feature vector space is characterized by drawing samples from the posterior distribution in a process of statistical inference.

The approximate Bayesian computation is a relatively new technique developed to allow a posterior distribution to be estimated without knowledge of the likelihood function [28,31]. The likelihood function measures the accuracy with which a model describes a set of data or, in this case, a measurement field

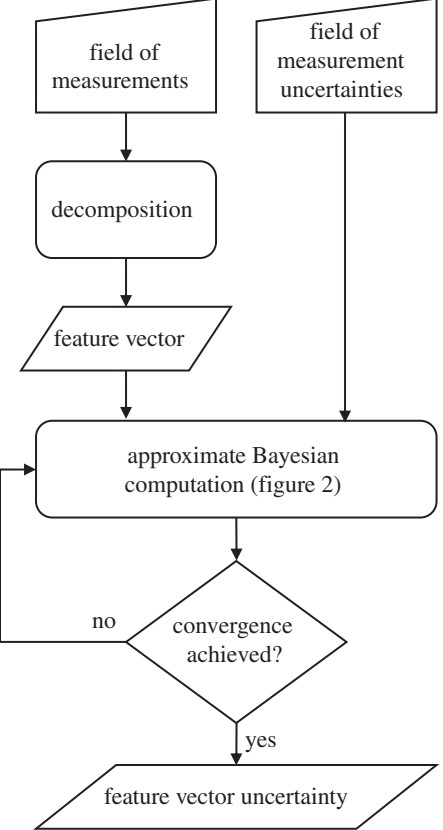

**Figure 1.** Flowchart for estimating the uncertainty in a feature vector representing a field of measurements for which the measurement error is known but without making any assumptions about the probability distribution of the measurement uncertainty.

being accurately represented by a set of coefficients in a feature vector. In a Bayesian analysis, a likelihood function is used to update a prior distribution to generate a posterior distribution, i.e.

$$p(\theta|D) = \frac{p(D|\theta)p(\theta)}{p(D|\theta)p(\theta)\,\mathrm{d}\theta'} \tag{2.1}$$

where $p(\theta|D)$ is the posterior distribution given the data $D$, $p(D|\theta)$ is the likelihood function and $p(\theta)$ is the prior distribution, while the denominator $p(D|\theta)p(\theta)\,\mathrm{d}\theta$ is a normalizing factor. Prior refers to the probability distribution that is assumed to reflect any previous knowledge about the variable or process being modelled, while posterior refers to the probability distribution updated based on some evidence. In the cases considered later, the prior distribution is the probability distribution of the feature vector describing the field of measurements and the posterior is the same distribution updated with information about the measurement uncertainty in the measurement domain. The prior is assumed to be a uniform distribution, defined over the range $[\theta_k \pm (2 \times \max(|M|))]$ for each coefficient of the feature vector, $\theta_k$, where $\max(|M|)$ is the magnitude of the maximum coefficient of the $\theta_k$, representing the field of measurements. This means that the prior distribution is centred on each coefficient and its width is equivalent to double the magnitude of the maximum coefficient. This selection was motivated by the fact that the largest portion of the variance within a dataset, which is related to the magnitude of the coefficients of the feature vector, is usually described by a few coefficients in a decaying manner, i.e. a few coefficients make up most of the variance, while the rest quickly decay to zero. This, combined with the fact that the measurement uncertainty is usually smaller than the variance within a dataset, allows the construction of this interval. Thus, the width of $2 \times \max(|M|)$ was selected as a rule of thumb. The likelihood function is undefined because the physical processes, by which the measurements are generated, are unknown, and in practice, it is usually unviable to quantify the probability distribution of the measurement error.

To circumvent the lack of information about the distribution of the measurement error, random samples from the posterior distribution are generated, using a Markov chain Monte Carlo (MCMC)

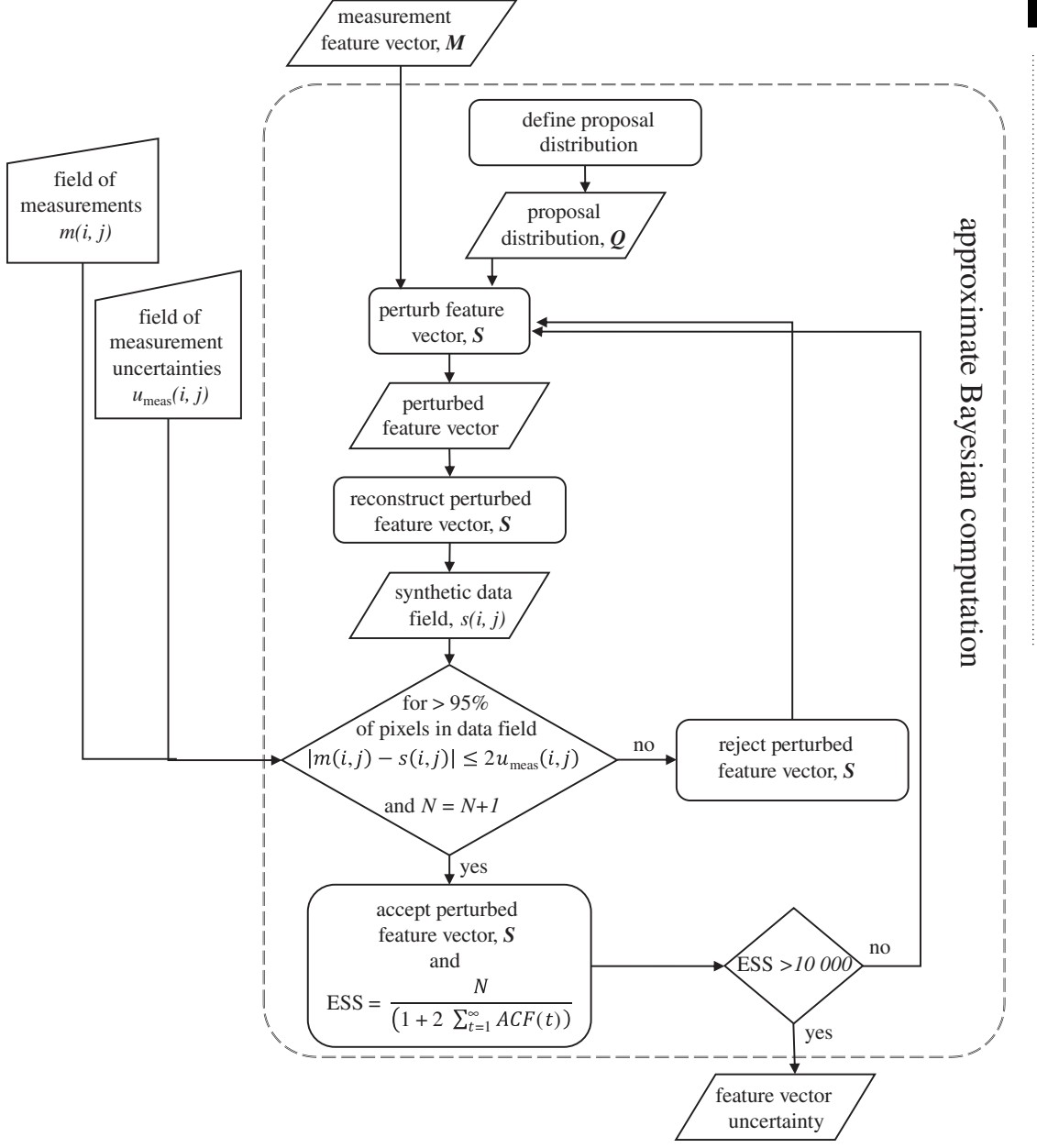

**Figure 2.** Sub flowchart illustrating the detail of the approximate Bayesian computation process shown in figure 1.

technique, and are compared with the experimental measurements using a distance measure to quantify the difference. A sample is accepted if the difference is less than or equal to the expanded uncertainty in the measurements, $2u_{meas}$. This process is repeated until sufficiently many acceptable samples have been generated to define the posterior distribution, i.e. the measurement uncertainty in the feature vector space. The process is summarized in the flowchart in figure 2. The implemented version of the approximate Bayesian computation uses the adaptive Metropolis algorithm [32] as its search tool in the feature space to iteratively search for feature vectors that, when reconstructed into measurement space, yield synthetic data fields for which less than 5% of the pixels deviate from the measurement data field by less than the measurement uncertainty in this space, i.e.

$$\text{at least } 95\% \text{ of } s(i,j) \text{ conform to: } |m(i,j) - s(i,j)| \leq 2u_{meas}(i,j), \qquad (2.2)$$

where $m(i,j)$ and $s(i,j)$ are the values of the measured and synthetic fields, respectively. The uncertainty in the measured values is allowed to vary spatially within the field of measurements by expressing it as $u_{meas}(i,j)$, and as in the CEN guide [27], the expanded uncertainty, $2u$, is used based on the Guide to the

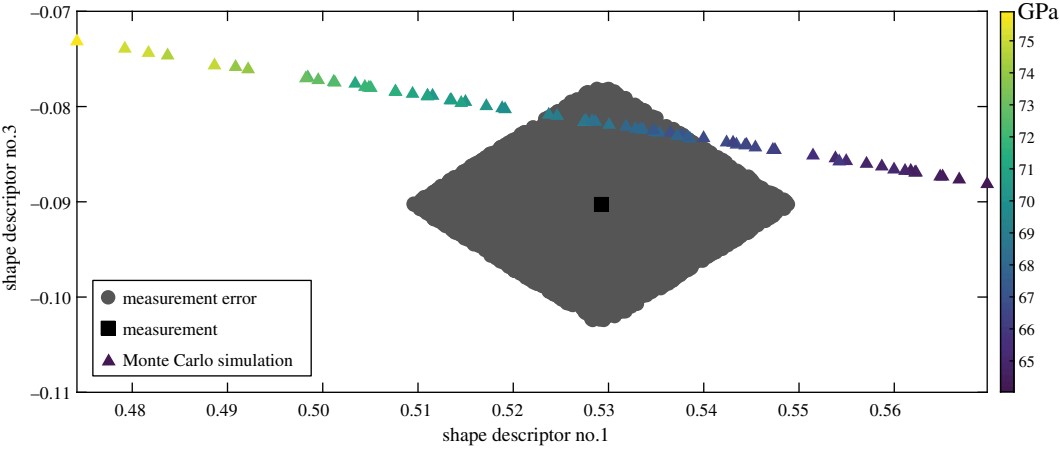

**Figure 3.** The clustered grey circles correspond to samples drawn from the posterior distribution during the approximate Bayesian computation and represent the measurement and its uncertainty in the feature vector space for the *y*-displacement field in the region of interest on the left side of the beam shown in figure 4. The measurement is described by a black square in the middle, while the results of a Monte Carlo simulation are shown using triangles whose colouring is based on the range of values of Young's modulus indicated by the colour bar.

Expression of Uncertainty in Measurement definition [24]. It should be noted that equation (2.2) would need to be modified in the event that the measurement error at each pixel location was very skewed.

The synthetic data fields are generated by perturbing the feature vector, representing the field of measurements. The perturbation is based on a proposal distribution $Q$ that is a multivariate Gaussian distribution. The choice of the proposal distribution is critical to achieving convergence to the posterior distribution efficiently because proposals must be neither too close to, and hence highly dependent on each other, nor too far apart, so that the synthetic values become unrepresentative of the measurement field. In addition, the proposal distribution affects the starting point for the search and the convergence rate. Hence, the standard deviation of the marginals of the proposal distribution is set initially at one-fifth of the absolute value of the maximum coefficient in the feature vector representing the measurement field, max($|\boldsymbol{M}|$), with a covariance of zero. After 1000 iterations, the covariance matrix is updated using data from these iterations, which makes the process more efficient [32]. In the event that after these initial iterations, the algorithm fails to find any perturbed feature vectors that are acceptable, then the standard deviation of the marginals in the proposal distribution is reduced until progress towards convergence is observed. There are various measures that can be used to assess the convergence of a MCMC method to a stationary posterior distribution. These include, among others, the Gelman–Rubin statistic [33], the Brooks–Gelman–Rubin statistic [34] and the effective sample size (ESS) [35]. The latter was used throughout the examples described in this article and is defined as follows:

$$\text{ESS} = \frac{N}{(1 + 2\sum_{t=1}^{\infty} \text{ACF}(t))}, \tag{2.3}$$

where $N$ corresponds to the number of iterations within each Markov chain, while $t$ represents the time lag used for the calculation of the autocorrelation function, ACF. A rule of thumb suggested by Kruschke [36] is an ESS of 10 000. This makes sure that at least 10 000 of the total iterations, $N$, are independent and representative values of the posterior distribution. The search was conducted three times for each dataset starting from different random starting points to ensure that the results are independent of the starting point [36,37].

Those perturbed feature vectors that satisfy the condition in equation (2.2) represent a distribution in feature space that characterizes the measurement uncertainty in that space. If the feature space is two dimensional, as in one of the datasets in the first example, where two components suffice to accurately represent the displacement measurements, then the samples of the posterior distribution can be plotted on a simple graph, as shown in figure 3; however, for a multicomponent space, a graphical representation is problematic. Nevertheless, scatterplot combinations can be used as an aid to visualize the drawn samples in a multicomponent space and can be equally employed to make

decisions about the similarity between datasets in model validation, updating or to identify changes in the condition of a system.

# 3. Applications

## 3.1 Bending displacements in a structural beam [18]

The first example is a simple I-beam with a series of holes in its web (the vertical slender section of the beam) and subject to three-point bending [18]. Two regions of interest have been selected in the web of the beam. In the first, the displacements can be described by a feature vector containing only two components, which renders the explanation of the method and graphical presentation of the results relatively straightforward. In the second region of interest, nine components were required in the feature vector to achieve an acceptable representation of the displacement field. The data were obtained by Lampeas *et al*. [18] based on a test designed as part of an inter-laboratory study [38]. As shown in figure 4, an aluminium I-beam of length 0.5 m and overall cross-section 42 × 65 mm with flanges (the two horizontal parts of the I) and web (vertical part of the I) of thickness 2.5 mm rested centrally on two supports that were 450 mm apart. The beam was loaded by moving the supports upwards, so that contact occurred between a loading nose situated at the mid-point on the top of the beam. A speckle pattern had been spray-painted onto the web of the beam, which allowed the displacements of the surface of the web to be tracked in three dimensions using stereoscopic images acquired using a pair of charge-coupled device cameras in a commercially available digital image correlation system (Aramis 5 M, GOM GmbH, Braunschweig, Germany). In this example, measurements of the displacement in the $y$-direction, i.e. the direction of the applied load, for the two regions of interest shown in figure 4, were used. Lampeas *et al*. [18] found the minimum measurement uncertainty to be 0.01 mm using a calibration procedure recommended by the CEN guide [27] that assumed it was spatially constant throughout the field of measurements.

In the first two examples, i.e. the structural beam and the soil moisture data, the fields of measurements were decomposed into feature vector space by fitting a set of orthogonal polynomials to the field and forming a vector from the resultant coefficients of the polynomials. A number of suitable types of polynomials are available, including Hahn, Krawtchouk, Legendre and Zernike, each exhibiting a unique set of characteristics among which is the sensitivity to local or global features, the type of the domain onto which the data are defined (polar or Cartesian) and the form of the polynomials (continuous or discrete) used [15]. Chebyshev polynomials have been used extensively across engineering and were adopted here; in part, because the decomposition process could be implemented using downloadable software that had been prepared for the inter-laboratory study [38] and was readily available [39]. The decomposition process was initially performed using Chebyshev polynomials with a very large number of coefficients. The CEN guide [27] for the validation of computational solid mechanics models recommends that the goodness of fit of the reconstruction of a data field to the original field should be assessed using the root mean squared residual, and this measure should not be greater than the measurement uncertainty $u_{meas}$ obtained from a calibration of the measurement system. It recommends that there should be no clusters of residuals greater than three times the root mean squared residual, where a cluster is defined as a group of adjacent pixels comprising 0.3% or more of the total number of values in the data field. In this example, for the first region of interest, towards the end of the beam, only the first and third shape descriptors had values that were substantially non-zero, and the reconstruction using only two coefficients gave an average residual that satisfied the conditions recommended in the CEN guide.

The approximate Bayesian computation, described in the flowchart shown in figure 2, was implemented in a specially written algorithm in MATLAB, which was based on one provided by Picchini [40]. The prior was a uniform distribution centred on the feature vector representing the measurement field with a half-width equal to two times the magnitude of the largest coefficient. The results obtained using the flowchart shown in figure 2 can be seen in figure 3, which shows the overlapping drawn samples (grey circles) representing the perturbed feature vectors whose reconstructed data fields are different from the measurement field by less than the expanded measurement uncertainty. The feature vector corresponding to the measured data field is shown as a black square and the resulting grey area represents the uncertainty defined by the measurement error. Figure 5 provides evidence of the convergence of the search algorithm to a stationary bivariate posterior distribution with traces of the values of the shape descriptors that the algorithm explored

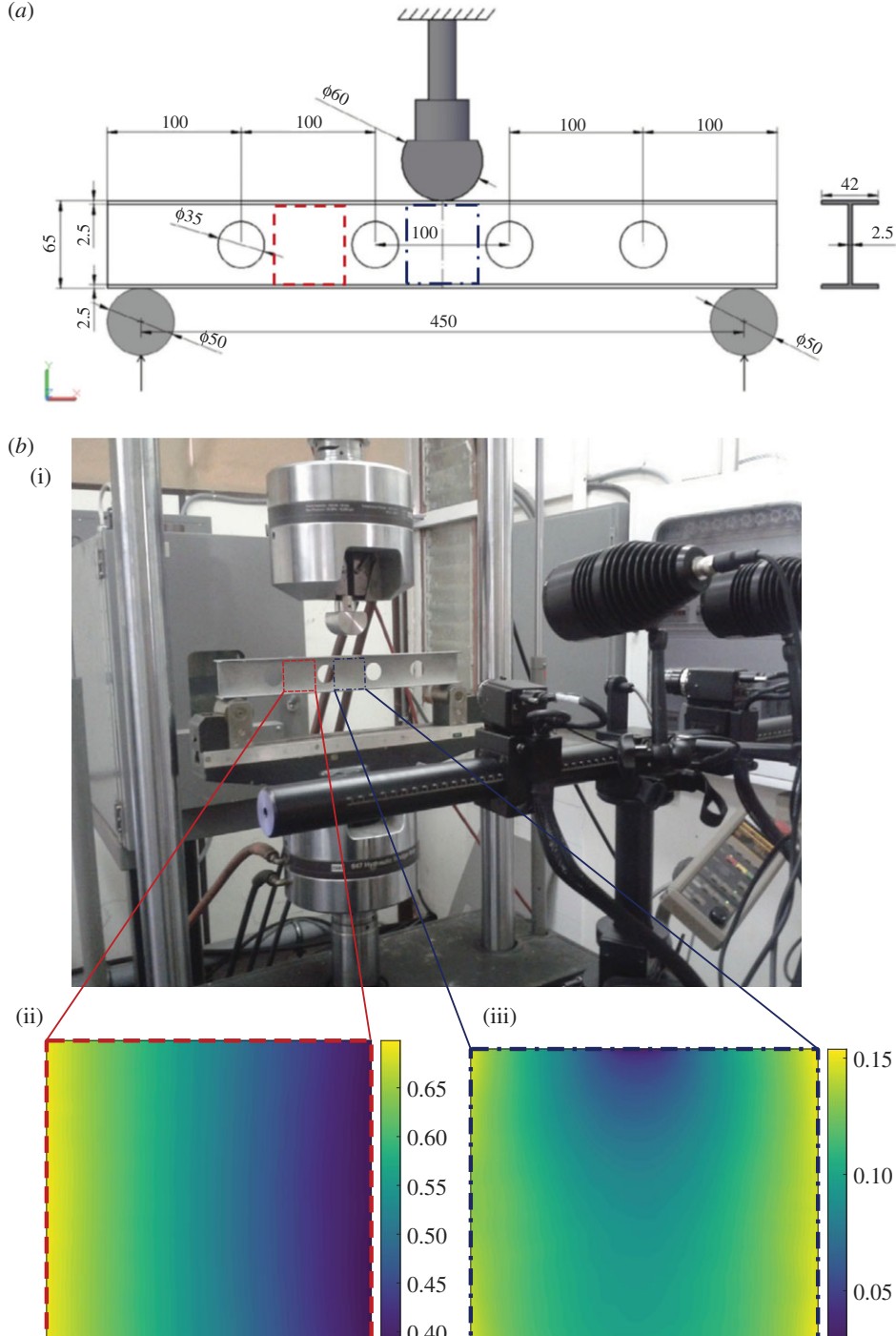

**Figure 4.** Experimental details for the first example showing the geometry and loading arrangements for the I-beam (*a*); the measurement set-up with the digital image correlation system in the foreground (*b*); and the vertical (*y*-direction) displacements of the two regions of interest in the web (adapted from the study by Lampeas *et al.* [18]).

and accepted during the search, the corresponding autocorrelations that should be close to zero, and the frequency distribution for the occurrence of each value of the shape descriptor during the approximate Bayesian computation, which is known as the posterior marginal distribution. The rapid decay of the autocorrelation function to zero demonstrates that each step was independent of its predecessor, which allowed convergence to be achieved quickly.

The second region of interest was in the web directly under the loading nose. This is a more complicated dataset shown at the bottom right of figure 4, which required decomposition using nine shape descriptors to satisfy the reconstruction criterion specified in the CEN guide. However, the

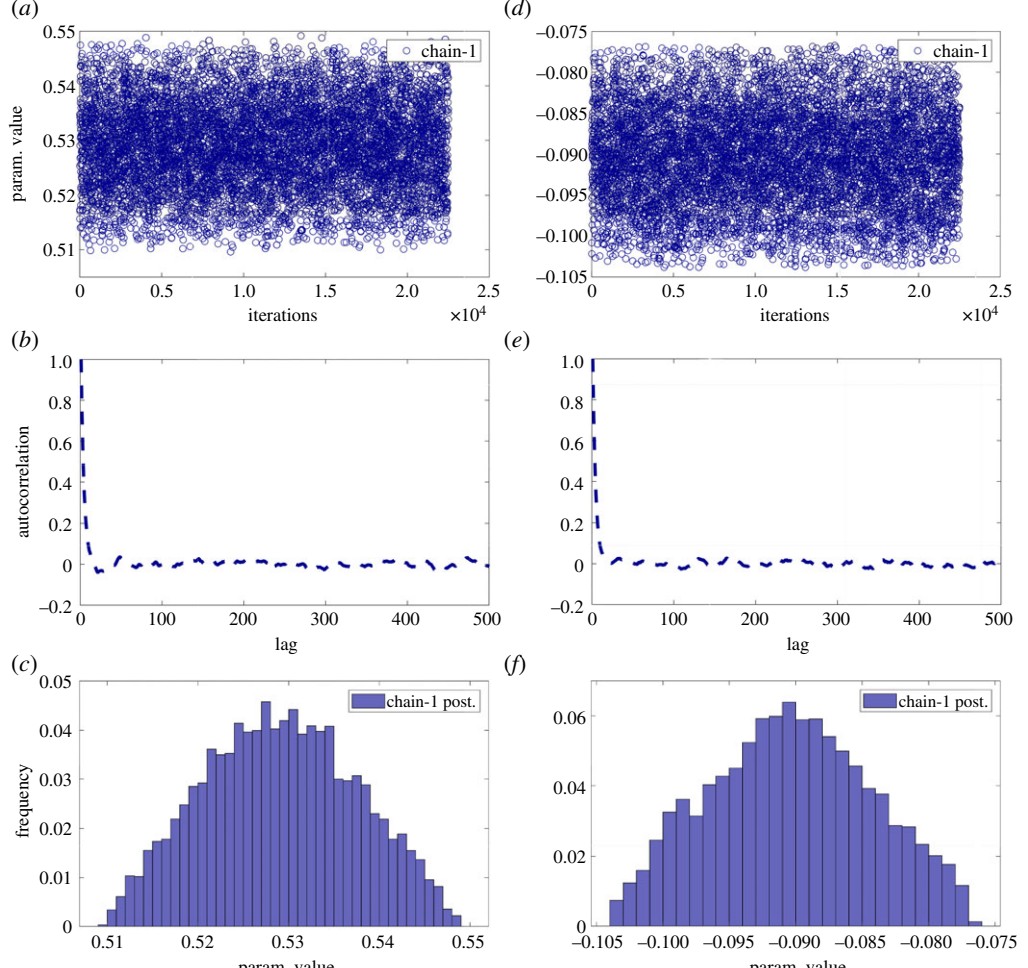

**Figure 5.** Evidence of convergence to the posterior distribution of the algorithm in figure 2 for shape descriptors no. 1 (a,b,c) and no. 3 (d,e,f) shown in figure 3. The path followed by the search is shown in the top graphs, the autocorrelation (middle) and the posterior distributions (bottom).

samples of only the three shape descriptors with largest magnitude, namely numbers 1, 6 and 2, drawn during the approximate Bayesian computation are shown in figure 6. Following the same convention as shown in figure 3, the measurement along with its uncertainty is shown in this three-dimensional plot. The feature vectors are shown that the algorithm visited and found their reconstructed data fields to be different from the measurement field by less than the expanded uncertainty, i.e. satisfying equation (2.2).

## 3.2 Moisture measurements at the Heihe River Basin [19]

Soil moisture data were used for the second example in which the measurement uncertainty varied spatially. The data for soil moisture shown in the top left of figure 7 represent the results of a Kriging analysis based on measurements from a wireless network of 162 ecological and hydrological sensors arranged non-uniformly in the Heihe River Basin in China [19]. Three different types of sensors with different measurement errors were used in the study. The variances from the Kriging analysis are shown in the top right of figure 7 and account for the sparsity of sensors and the heterogeneous measurement error. The proposed method combining orthogonal decomposition and the approximate Bayesian computation was implemented using the dataset on the top left as the measured quantity and the dataset on the top right as the field of uncertainties. Owing to the complexity of the measurement field, the initial decomposition was performed using 1000 coefficients in the Chebyshev polynomials, and then, the 100 largest non-zero coefficients were retained as elements in the feature vector to satisfy the requirements for quality of the representation recommended in the CEN guide [27].

An unavoidable consequence of the complex shape of the data field is the large number of shape descriptors required in the feature vector to represent it to the required accuracy. However, it has been

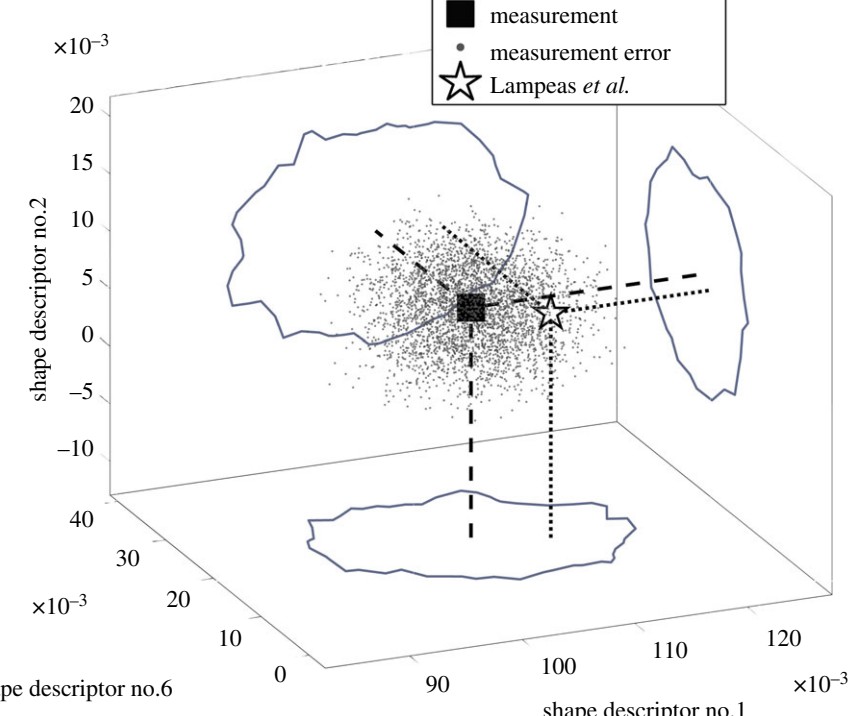

**Figure 6.** Uncertainty bounds for the $y$-direction displacement in the region of interest directly under the loading nose in the I-beam shown in figure 4 based on the three shape descriptors with the largest magnitude that represent 99% of the total variability in the measurement data. The 'cloud' of points represents the samples drawn from the posterior distribution, obtained using the flowchart in figure 2 and corresponds to the measurement and its uncertainty. The prediction by Lampeas *et al.* [18] lies within the cloud as demonstrated by its projection and the outline of the convex hull enclosing the points.

shown that the adaptive Metropolis algorithm, used in approximate Bayesian computation, can efficiently handle searches in such high dimensions [32]. Some of the results of the search are shown at the bottom of figure 7 for combinations of the five of most significant shape descriptors. The array of plots represents an attempt to present the five-component samples of points that characterize the uncertainty for the measurement field in the low-dimension or feature vector space. This multivariate 'cloud' of points corresponding to the samples drawn from the posterior distribution could be used to assess, for example, the significance of changes in the soil moisture over a time period.

## 3.3 Monthly oceanographic temperature fields [20,21]

In 2000, a global network, currently consisting of about 3800 Argo profiling floats, was established with the aim of systematically observing the temperature and salinity of the world's oceans. The resultant high-quality and spatially dense data have allowed researchers to obtain a better image of the properties of the world's oceans and their interaction with climate changes. The information gained from the network is being used to drive policy changes related to climate and also to validate climate models [41].

However, before the data can be used for climate research, quality flags are attributed to the measured quantities and those that pass the quality requirements are assembled through a process of optimal interpolation into plotted fields, such as the one shown in figure 8, which is based on the In Situ Analysis System 13 for which more details can be found in reports by Gaillard and his co-workers [20,21]. The results of this interpolation process are monthly averaged temperature and salinity fields across the globe along with fields of errors that are based on four components: (i) the measurement error of the floats, (ii) the variance in these fields measured within a time frame of 41 days with respect to the mean, (iii) the uncertainty arising from the interpolation process, and (iv) previous statistical knowledge in parts of the ocean where measurements are scarce and estimates are provided by previous analyses.

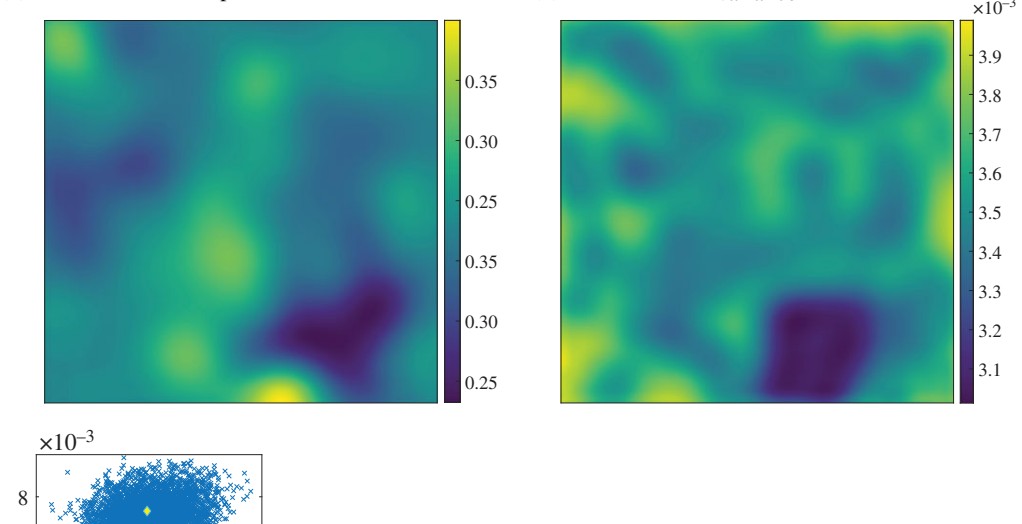

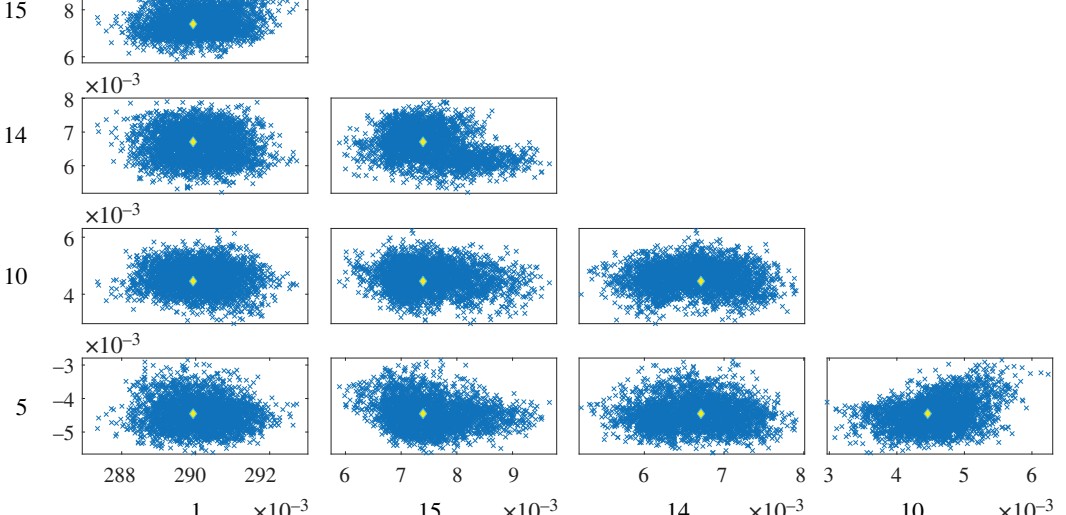

**Figure 7.** Spatial distribution of soil moisture data from the Heihe River Basin digitized using data from Kang *et al.* [19] based on the results of Kriging interpolation (*a*) from sparse measurement locations with heterogeneous measurement errors represented by the Kriging variance (*b*); and the corresponding uncertainty bounds (*c*), based on the five most significant shape descriptors. The measurement field is described by a feature vector represented by the yellow diamond.

The data fields used in this example are monthly temperature data spanning a total of 11 years from 2002 to 2012 from Gaillard [21], and the illustrative data in figure 8 are for September 2007. PCA was used to decompose the monthly temperature data. PCA allows the projection of high-dimensional data into a lower dimensional space by retaining only the coefficients of the components that account for the largest percentage of the total variability in the data [30,42]. This results in a set of uncorrelated orthogonal basis vectors, each representing a certain feature or mode of the dataset and a set of coefficients. The dataset can then be reconstructed as a linear combination of the basis vectors and the corresponding coefficients, i.e. the outcome is similar to decomposition using orthogonal polynomials though the process is different with the result that features or modes are dependent on the form of the original dataset, whereas they are fixed in the polynomial decomposition. The analysis involved a number of steps: initially, the 132 monthly temperature fields, consisting of the same-sized matrices for each month, were reshaped into vectors, after the gaps in the datasets representing land masses were removed leaving 270 733 values in each vector. Second, these vectors were assembled into a large 270 733 × 132 matrix in which each vector formed a column. Third, the matrix was centred around its mean and decomposed using PCA to generate a matrix of coefficients of the principal components, with a feature vector for each month and a matrix of eigenvectors corresponding to the principal components. The complexity of the ocean temperature distributions required 100 principal components to describe them, so that the root mean square error of each reconstructed dataset compared to its corresponding original dataset was always less than the mean uncertainty in the temperature measurements.

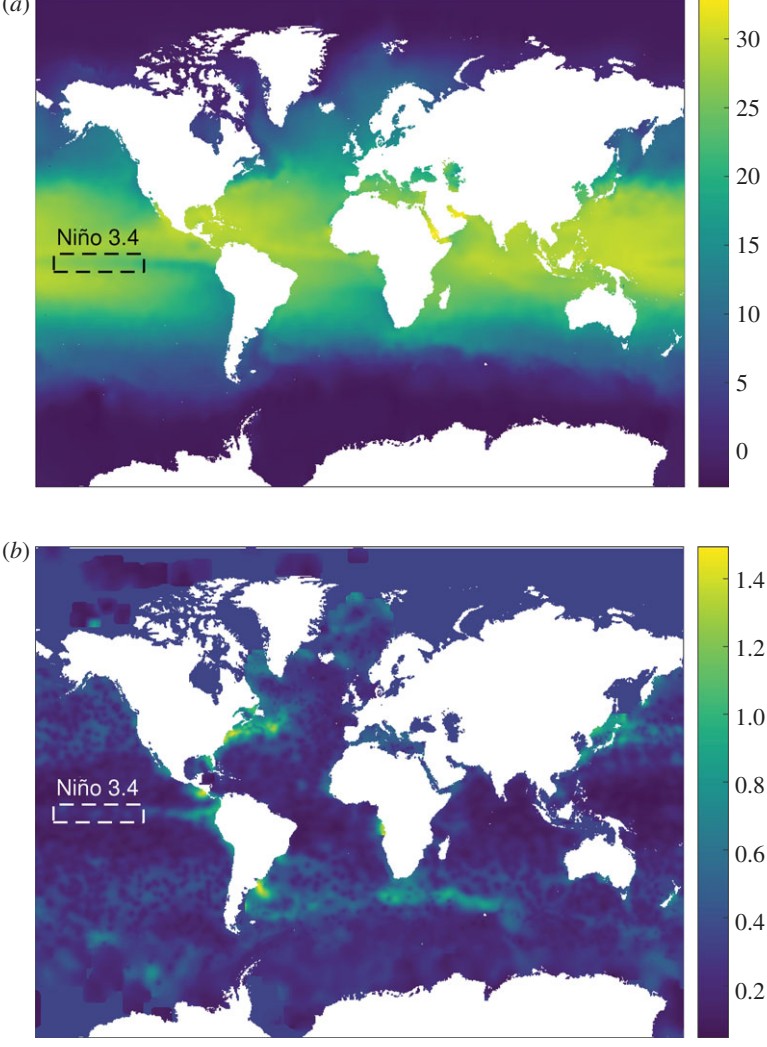

**Figure 8.** Monthly ocean temperature (°C) distribution (*a*) at a depth of 10 m for September 2007 and corresponding error field (°C) (*b*) from Gaillard [21]. Niño region 3.4 is shown in the dashed rectangles.

Finally, approximate Bayesian computation (figure 2) was performed using the feature vectors and the monthly fields of uncertainties as inputs, while the principal components were used for the reconstruction of the perturbed feature vectors. The prior was a uniform distribution centred on the feature vector representing the measurement with a half-width equal to twice the absolute value of the first principal component to minimize the effect of the prior on the posterior distribution. The results are shown in figure 9 for the 10 most significant principal components for September 2007 using temperature and error fields from an ocean depth of 10 m. The cloud of points represents the uncertainty bounds on the temperature data in the feature vector space and can be used to evaluate the significance of trends in the data in this space.

# 4. Discussion

## 4.1. Representing the measurement error

The objective of this study is the development of a method to characterize the uncertainty associated with spatial measurements in a low-dimensional form without making any assumptions about the probability distribution of the measurement error. Various types of mathematical transformations can be used to extract features or patterns from the data to reduce its dimensionality [4,15]. In various applications where decisions need to be made about whether two states are equivalent, for example, in condition monitoring or model validation, when the associated uncertainties are not negligible and the decision

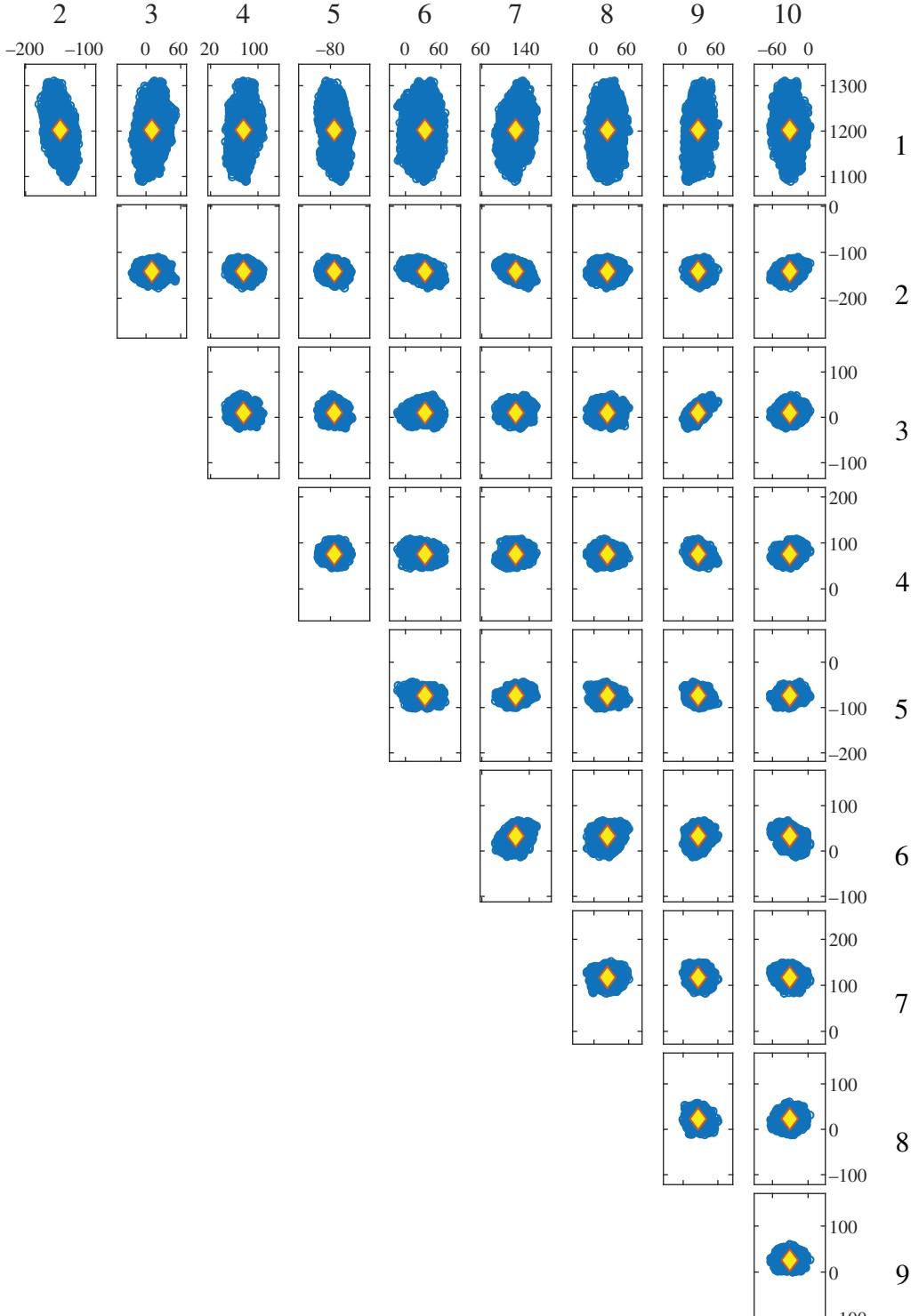

**Figure 9.** The distribution of measurement error in the feature vector space for the first 10 principal components for the oceanographic data in figure 8. The measurement is depicted by the diamond at the centre of each plot.

is to be based on a representation of the data in a lower dimensional form, then it is important to be able to assess whether a pair of feature vectors belongs to the same population, and this assessment should be made by evaluating the difference in corresponding shape descriptors using the associated uncertainty for the shape descriptor. In this article, it has been proposed that the extent of the uncertainty bounds in feature vector space can be established by using approximate Bayesian computation via an adaptive Metropolis algorithm to search for perturbed feature vectors that, when reconstructed into measurement space, generate synthetic data fields that deviate by less than the expanded

measurement uncertainty from the measured data field. The set of such perturbed feature vectors represent the uncertainty bounds in the feature vector space. In a two-dimensional space, such as in the first region of interest in the first example, this set of perturbed feature vectors can be readily represented in a graph as shown in figure 3; however, when the feature vector space involves many components, then sets of scatterplots, such as those shown in figure 9, can be used to represent the multicomponent uncertainty.

For the cases where normality is reasonably assumed, the practitioner could modify the proposed method and employ the spatial data analysis using Kriging with homogeneous or heterogeneous measurement errors [43] to obtain estimates for the true underlying data field. Then, these estimates could be decomposed into the low-dimensional space and would represent the measured data with its associated measurement uncertainty. However, this approach involves additional complexities caused by the need to model the spatial covariance of the data and more assumptions associated with process stationarity [44], which make it less attractive as a black-box approximation of the low-dimensional error. Similarly, with knowledge of the probability distribution of the measurement error, it might be possible to use a parametric bootstrap to sample the distribution multiple times and decompose the samples to create its representation in feature vector space. However, the method proposed here removes the need to make any assumption about distribution of the measurement error. A nonparametric bootstrap could also be considered although the difficulty in establishing the covariance matrix when there are thousands of point measurements is likely to make such an approach prohibitive.

## 4.2. Validation

One of the potential applications of the proposed technique is in model validation. Decision makers want to know whether they can trust the predictions made by models across science and engineering, from mechanics and meteorology to climate modelling and finance. Part of the process of establishing trustworthiness is to perform a validation process, which has been defined as 'determining the degree to which a model is an accurate representation of the real world from the perspective of the intended uses of the model' [45, p. 3]. This is not straightforward when fields of measurements and predictions are available, particularly when the data fields have different grid densities, orientation and scales. Thus, to alleviate these issues in structural mechanics, the CEN guide for validation of computational models in solid mechanics [27] recommends reducing the dimensionality of the data fields using orthogonal decomposition by employing suitable polynomials, as described for the first example. However, once the data are reduced to a lower dimensional space, a rigorous representation of the associated measurement uncertainty is seldom made. Instead, the CEN guide recommends plotting the shape descriptors, describing the measured and predicted data fields, against one another and assessing whether the resultant points lie within an uncertainty interval defined by

$$s_p = s_m \pm 2u(s_m), \tag{4.1}$$

where $s_p$ and $s_m$ are the shape descriptors representing the predictions and measurements, respectively, and $2u(s_m)$ is the expanded uncertainty in the shape descriptor describing the measurements, which is given by

$$u(s_m) = \sqrt{u_{\text{meas}}^2 + u^2}. \tag{4.2}$$

$u_{\text{meas}}$ is the measurement uncertainty obtained from a calibration of the measurement instrument, while $u$ is the average residual obtained from comparing the reconstructed and original data fields as mentioned earlier. In the process described in the CEN guide, it is assumed that the measurement uncertainty is uniform over the field of measurements and is not transformed into the shape descriptor space. However, the proposed methodology allows a graphical representation of the measurement uncertainty in the low-dimensional space as a cloud of points with the shape descriptor representing the measurement values at its centre, as shown in figure 3 for the region of interest towards the end of the I-beam. In addition, when performing an experiment with the I-beam, as shown in figure 4, Lampeas *et al.* [18] also predicted the behaviour of the beam using a finite-element model. The predicted field of displacements for the region of interest in the centre of the beam was decomposed using Chebyshev polynomials in exactly the same way as the measured field and the resultant shape descriptors are plotted in figure 6 and lie just within the cloud of points, representing the uncertainty interval for the measurements. Thus, it could be concluded that the model is an

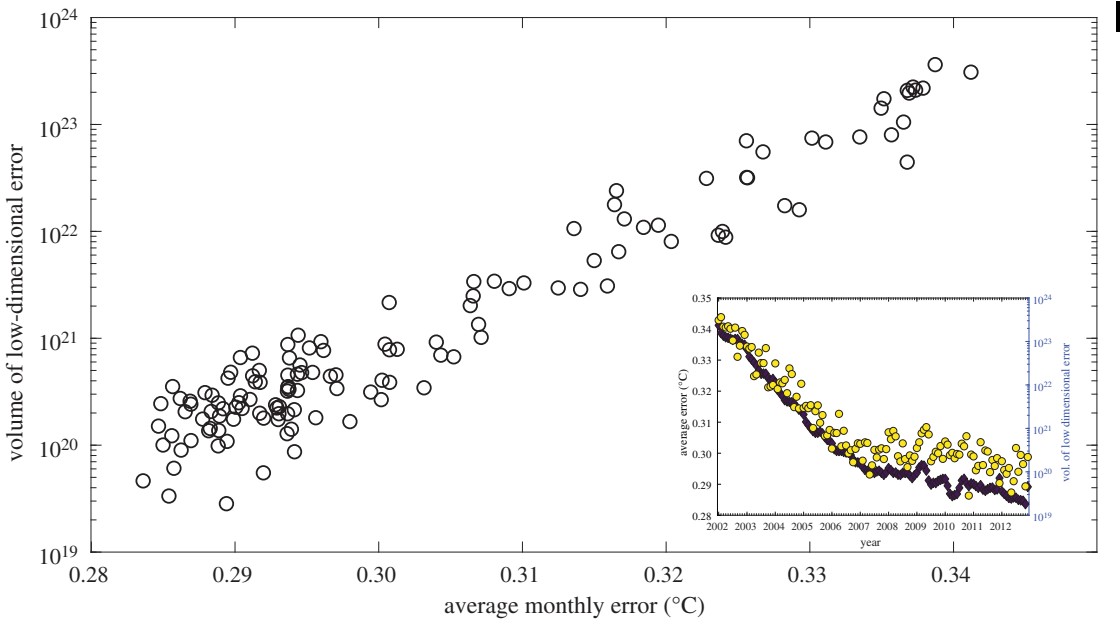

**Figure 10.** The volume of the cloud of points representing the posterior distribution as a function of the monthly average errors, i.e. the spatial average of the errors in the dataset for each month, for the monthly ocean temperature data from Gaillard [21]; and inset, average error (diamonds) and the volume plotted as a function of time.

acceptable representation of the experiment because the difference between the predictions and measurements is less than the expanded uncertainty in the feature vector space, following the same principles as the CEN guide but applying them completely in the feature vector space. This conclusion agrees with the one that was drawn by Lampeas *et al.* [18] using the criterion described by equations (4.1) and (4.2).

## 4.3. Calibration

It is also possible to use the uncertainty described by the posterior distribution to calibrate a model. For instance, when the finite-element modelling for the I-beam is repeated using a series of values for the Young's modulus varying between 65 and 75 GPa, then the series of coloured triangles in figure 3 represent the predicted displacement fields. Because the values for the Young's modulus that yield shape descriptors within the distribution lie between 67 and 71 GPa, it can be concluded that this range would be acceptable when considering the displacements in this region of interest.

Although it is relatively straightforward to develop a computational model of the structural behaviour of the beam in the first example, it is considerably more complicated to construct computational models for soil moisture or for ocean temperatures owing to the large number of parameters involved and the complexity of the interactions between factors influencing the responses. In such circumstances, it is often impractical to perform multiple runs of a model; thus, an alternative is to employ techniques such as meta-modelling to overcome the issue. Meta-models are surrogates for models of the system of interest in which the relationship between the inputs and outputs of the original model is represented mathematically using a technique such as an artificial neural network, polynomial chaos expansion or Gaussian process regression. These techniques can successfully describe the complex mapping between outputs and inputs; however, they do not provide a representation of the associated uncertainty in the corresponding space. The proposed methodology could be used alongside such techniques to accurately represent the uncertainties in the reduced-order or feature vector space. This is effectively the process represented by the example using the soil moisture data from the Heihe River Basin [19] that are based on the results of Kriging interpolation.

## 4.4. Identification of critical changes

It would be expected that the volume of the cloud of points characterizing the measurement uncertainty in feature space would be correlated with other measures of the errors in the measurements. This has

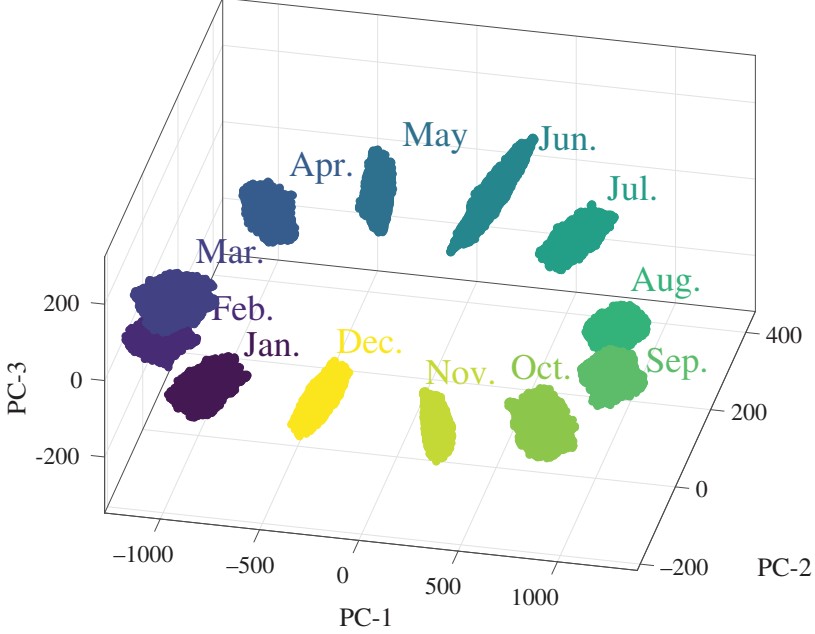

**Figure 11.** Convex hulls fitted to the cloud of points representing the uncertainty intervals for the ocean temperature measurements for each month in 2002 using only the three most significant principal components (PCs). The lack of overlap between hulls can be interpreted as implying a significant difference in the temperature between months.

been shown in figure 10 for the ocean temperature data by plotting the volume of the cloud of points representing the posterior distribution as a function of the monthly average errors, i.e. the spatial average of the errors in the dataset for each month. The volume was calculated as the square root of the determinant of the covariance matrix of the distribution. The calculation of the determinant of the covariance matrix as an estimate of the scatter of a multivariate distribution has been reported in various sources such as [46] and [47]. A covariance matrix consisting of 10 of the most significant principal components was used in this case and gave a correlation of 0.975 with the monthly average errors; the noise in the volume data is probably a result of the complexity associated with the dimensionality of the problem. The need to characterize the temporally varying uncertainties in measurements could be important as new equipment may be added to enhance the overall credibility of the measurements or to remove damaged sensors.

Convex hulls were fitted to the volume defined by the 'cloud of sampled points' for the temperature data for each month in 2002 and are shown in figure 11. It can be seen that they are distributed along an approximately elliptical path running clockwise through the year. There is no overlap between the hulls, which implies it would be reasonable to conclude that there is a significant difference between the global temperature pattern in each month. A more sophisticated analysis is possible by examining the behaviour of certain shape descriptors or principal components. For instance, it was observed that the fifth principal component (PC-5) describing the monthly distribution of temperature could be used to characterize the El-Niño Southern Oscillation (ENSO) as shown in figure 12. The ENSO is an irregular cycle of recurring warm (El Niño) and cool (La Niña) patterns of temperature in the tropical Pacific that occur every 2–7 years and cause major disruptions in the climate [49]. The Oceanic Niño Index (ONI) is the difference between the 3-month average and the 30-year average of the surface temperature of the ocean in an area of the east-central tropical Pacific between 5° N and 5° S and between 120° and 170° W, which is known as the Niño 3.4 region and is shown in figure 8 [50]. The correlation between the value of the PC-5 and the ONI was 0.88, which implies that PC-5 captures the characteristics of the ENSO phenomenon. The methodology proposed in this study can be used to define an uncertainty interval for each principal component as the distance across the cloud of points in the direction corresponding to each component. It was found that this uncertainty interval can be used as an indicator of an ENSO phenomenon when the value of PC-5 varies from zero by more than its expanded uncertainty for three consecutive months. The variation of the value of PC-5 and its uncertainty interval are plotted in figure 12 as a function of time, together with the value of the ONI [48], and the shape of PC-5 is shown as an inset. The Climate Prediction Center of the National Oceanic and Atmospheric Administration consider that La Niña conditions exist when the ONI is less than or equal to −0.5 and

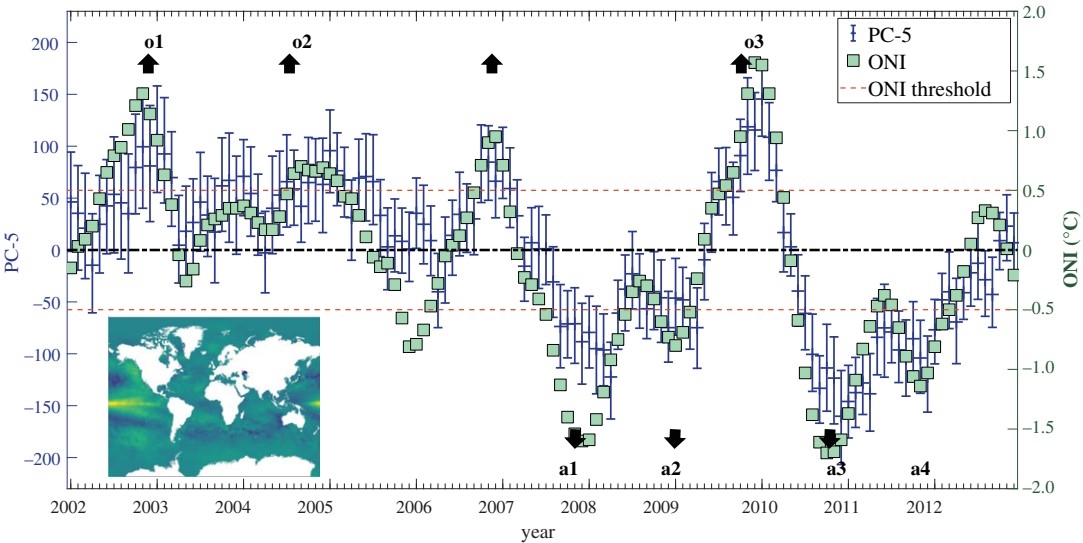

**Figure 12.** The magnitude of the fifth principal component, PC-5 of global ocean temperature at a depth of 10 m and the Oceanic Niño Index (from [48]) as a function of time. El Niño and La Niña phenomena based on the ONI are highlighted by o1 to o3 and a1 to a4, respectively; while the corresponding phenomena indicated by the PC-5 varying from zero by more than its uncertainty are indicated by upward and downward errors, respectively. The inset shows the shape of the fifth principal component.

El Niño conditions when it is greater than or equal to 0.5 for at least five consecutive months. La Niña and El Niño events identified using the ONI criteria are numbered in figure 12 using the prefix $a$ and $o$, respectively; while those identified using the PC-5 criteria are shown by upward and downward arrows, respectively. It can be seen that the PC-5 criterion predicts all of the ONI indicated events but also gives one false positive in the autumn of 2006 when the ONI is over the 0.5 threshold for only 4 months. In 2011–12, the PC-5 criterion indicates a 20-month La Niña event, whereas the ONI implies two events separated by 3 months. This proposed approach to classifying the occurrence of ENSO phenomena has a number of advantages over the ONI, namely, that the uncertainty interval could be used to provide a level of confidence in the classification; it should be more representative of the mechanisms driving the ENSO phenomena because it is based on the global pattern of ocean temperatures; and it should allow straightforward comparison of predictions and measurements while accounting for the global ocean dynamics.

## 4.5. Implementation

The reasons for selecting approximate Bayesian computation to search for the posterior distribution of the measurements in the coefficient or feature vector space were as follows: (i) its tractability, especially when moving to multivariate spaces compared to other techniques such as 'history matching' [51], which would require a much larger number of iterations; (ii) the rich literature around MCMC techniques; (iii) the potential capabilities for faster convergence using techniques such as adaptive stepping [32]; and (iv) and the ability to run multiple 'chains' of calculations independently, thus exploiting parallel programming capabilities in modern computing.

The principal benefit stemming from the application of the new methodology is the way in which it supplements existing techniques for reducing the dimensionality of information-rich data fields by allowing the associated uncertainty to be characterized and represented in the low-dimensional space. As the examples have demonstrated, the results from the methodology provide a visual and intuitive way to inform the decision makers about the variability in the data and the significance of the difference between data fields based on rigorous statistical principles. Compared to traditional geostatistical approaches to characterizing the uncertainty in spatial data, where a large number of assumptions and choices must be made during the analysis, the proposed methodology requires only two parameters to be selected, namely, the size of the uncertainty interval and the confidence level required (e.g. $2u(i, j)$ for a 95% confidence interval); hence, the methodology can be used as a 'black-box' approach. This is attributed to the fact that the primary spatial characteristics of the data are captured during the decomposition process and subsequently used to represent the associated uncertainty. The resulting distribution in a low-dimensional space, representing the spatial data fields, is easier to handle using multivariate statistics, thus allowing

inferences to be drawn. Finally, employing the proposed methodology allows all of the available spatial information to be included in the analysis. This is important in activities like model validation and model calibration or updating, where all the existing information, including both measurement values and the accompanying uncertainty, should be taken into consideration when making decisions.

# 5. Conclusion

A novel methodology has been developed that allows the transformation of measurement error into a characterization of the uncertainty of a feature vector representing a field of measurement values in a reduced-order space, without making any assumptions about the probability distribution of the measurement error. The method uses approximate Bayesian computation with an adaptive Metropolis algorithm to search for the posterior distribution of the measurement values and their uncertainty in the feature vector space. The result is a distribution in the feature vector space that characterizes the measurement and its uncertainty and forms a multivariate uncertainty estimate that can be used to evaluate the significance of differences between data fields.

There are three innovations in this methodology:

(i) its capability to characterize the uncertainty in the elements of a feature vector when the uncertainty in the underlying measurements is spatially constant or varying. This uncertainty may be obtained either from the calibration process for a device capable of measurements across a field of view or through statistical postprocessing as in the case of spatially dispersed sensors whose error is heterogeneous;

(ii) its applications to the validation or confirmation of models in engineering mechanics or 'forecast verification' of meteorological models where decisions regarding the capability of the model to represent the real-world must be made; and

(iii) its wide range of applications, extending from two-dimensional data fields from tests on engineering structures to three-dimensional data fields for which volumetric data are available relating spatially and temporally varying temperatures, where this methodology could be used to identify significant changes between measurements and predictions, or between successive measurements obtained over time indicating the change in condition of a system, such as the ENSO.

The proposed methodology supplements techniques for reducing the dimensionality of information-rich data fields by permitting the associated uncertainty in the data to be characterized and represented in the low-dimensional or feature vector space, without making any assumptions about the probability distribution of the measurement error. The examples presented show that the results from the proposed methodology can be presented in a visual and intuitive manner to inform decision makers about the uncertainty in data and the importance of differences between data fields, such as disagreements between measurements and predictions or changes in conditions over time, while allowing multivariate statistics to be used, so that inferences can be drawn.

Data accessibility. Data on which this study is based are available from the Dryad repository at: https://doi.org/10.5061/dryad.6hdr7sqx2 [52].

Authors' contributions. A.A. performed all of the data analysis and coding involved in the study; S.F. and E.A.P. conceived and supervised the work; all authors contributed to the final manuscript following preparation of a first draft by A.A. and E.A.P.

Competing interests. The authors have no competing interests.

Funding. A.A. was supported by an EPSRC CASE Award PhD (grant no. 13220001) sponsored by Airbus SAS (project id. GP/RA/1160).

Acknowledgements. The authors appreciated the helpful discussions with Eszter Szigeti, Linden Harris, and Sanjiv Sharma of Airbus.

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
