## [Peer Review File · Royal Society Open Science]

Review History

Decision letter (RSOS-200502.R0)

06-Apr-2020

Dear Mr Alexiadis,

Manuscript ID RSOS-200502 entitled "Transformation of measurement uncertainties into low-dimensional feature vector space" which you submitted to Royal Society Open Science, has been reviewed. The comments from reviewers are included at the bottom of this letter.

In view of the criticisms of the reviewers, the manuscript has been rejected in its current form. However, a new manuscript may be submitted which takes into consideration these comments.

Please note that resubmitting your manuscript does not guarantee eventual acceptance, and that your resubmission will be subject to peer review before a decision is made.

Once you have revised your manuscript, go to <https://mc.manuscriptcentral.com/rsos> and login

to your Author Center. Click on "Manuscripts with Decisions," and then click on "Create a Resubmission" located next to the manuscript number. Then, follow the steps for resubmitting your manuscript.

Your resubmitted manuscript should be submitted by 04-Oct-2020. If you are unable to submit by this date please contact the Editorial Office.

Kind regards,
Lianne Parkhouse
Royal Society Open Science
openscience@royalsociety.org

on behalf of Professor Len Thomas (Associate Editor) and R. Kerry Rowe (Subject Editor)
openscience@royalsociety.org

Associate Editor Comments to Author (Professor Len Thomas):

Thank-you for your submission. While the method you propose may be sound, I simply do not understand the reason why one may want to use it, compared to the much more simple alternative of a parametric bootstrap. As I understand your motivation, the goal is to transform measurement error, either in the form of known distributions or a set of bounds (measured value $\pm 2 \mu_{\text{meas}}$) coupled with a uniform distribution, into a characterization of uncertainty on a reduced feature space that is created by a deterministic set of operations. A straightforward way to achieve this is a parametric bootstrap - i.e., to sample multiple times from the measurement error distributions and for each sample repeat the deterministic operations to produce multiple realizations of the reduced feature space; the distribution of these realizations then forms a simulation-based approximation of the distribution of uncertainty in the output given the input measurement error distribution.

Given this, I am recommending rejection of your manuscript. I will, however, be happy to reconsider if you can provide a compelling justification of why the proposed method is superior to the one outlined above. Apologies if I have misconstrued some aspect of the problem being addressed.

With best wishes, Len Thomas

Author's Response to Decision Letter for (RSOS-200502.R0)

See Appendix A.

RSOS-201086.R0

Review form: Reviewer 1

Is the manuscript scientifically sound in its present form?

Yes

Are the interpretations and conclusions justified by the results?

No

Is the language acceptable?

Yes

Do you have any ethical concerns with this paper?

No

Have you any concerns about statistical analyses in this paper?

Yes

Recommendation?

Major revision is needed (please make suggestions in comments)

Comments to the Author(s)

The paper proposed an interesting method using approximate Bayesian computation, while the basic concept/idea of reducing the dimensionality of data matrices to feature vectors was published before. The paper mentioned "decision-making" a number of times, but it is not clear that what decision-making it can help, to what degree/ confidence level? the reviewer recommend to provide an example to elaborate further on a real decision-making case, how can this concept/method help to determine the consequence/risk of a decision-making using the low dimensional feature vector space?

In case a measurement uncertainty is uniform over the field of measurements, for example depending on stress gradient or surface finish, the error distribution is very skewed over the field, can the proposed concept and method still determine a reasonable error distribution?

Decision letter (RSOS-201086.R0)

Dear Mr Alexiadis

On behalf of the Editors, we are pleased to inform you that your Manuscript RSOS-201086 "Transformation of measurement uncertainties into low-dimensional feature vector space" has been accepted for publication in Royal Society Open Science subject to minor revision in accordance with the referees' reports. Please find the referees' comments along with any feedback from the Editors below my signature.

Please submit your revised manuscript and required files (see below) no later than 7 days from today's (ie 21-Jan-2021) date. Note: the ScholarOne system will 'lock' if submission of the revision

is attempted 7 or more days after the deadline. If you do not think you will be able to meet this deadline please contact the editorial office immediately.

on behalf of Professor Len Thomas (Associate Editor) and R. Kerry Rowe (Subject Editor)
openscience@royalsociety.org

Associate Editor Comments to Author (Professor Len Thomas):

Comments to the Author:

Please accept my apologies for the longer-than-usual time to provide a response on this manuscript. It has proven very difficult to obtain reviewers. We have received one review; the reviewer was broadly supportive of publication and I concur. Based on my reading of their comments I am recommending acceptance with minor revisions. In your resubmission, please provide a point-by-point response to their comments, and to mine below. I will review the changes and your responses and make a final decision – I do not think the manuscript will need to go for review again.

Regarding the reviewer's last comment "In case a measurement uncertainty is uniform ... the error distribution is skewed over the field" I'm afraid I cannot understand the comment. I hope you can, but in case you cannot either, just let me know in your response letter.

Additional points to respond to are:

1. Please ensure you provide all data and code required to reproduce the results in this paper, in line with the journal's policy. Failure to do this will result in rejection of the paper.
2. Page 11 "Similarly, with knowledge of the probability distribution of the measurement error, it might be possible to use a parametric bootstrap to sample the distribution..." Thank-you for mentioning this. If the sample size of measurements is sufficient, might it also be possible to use a non-parametric bootstrap, sampling the measurements with replacement, to quantify uncertainty?
3. Page 16, last sentence "... and the significance of differences between data fields..." What significance do you mean, exactly, here? Presumably not statistical significance in the classical hypothesis testing meaning. Assuming you mean "importance" can you please give an example?

Reviewer comments to Author:

Reviewer: 1

Comments to the Author(s)

The paper proposed an interesting method using approximate Bayesian computation, while the basic concept/idea of reducing the dimensionality of data matrices to feature vectors was

published before. The paper mentioned "decision-making" a number of times, but it is not clear that what decision-making it can help, to what degree/confidence level? the reviewer recommend to provide an example to elaborate further on a real decision-making case, how can this concept/method help to determine the consequence/risk of a decision-making using the low dimensional feature vector space?

In case a measurement uncertainty is uniform over the field of measurements, for example depending on stress gradient or surface finish, the error distribution is very skewed over the field, can the proposed concept and method still determine a reasonable error distribution?

===PREPARING YOUR MANUSCRIPT===

===PREPARING YOUR REVISION IN SCHOLARONE===

Please ensure that you include a summary of your paper at Step 2 'Type, Title, & Abstract'. This should be no more than 100 words to explain to a non-scientific audience the key findings of your

research. This will be included in a weekly highlights email circulated by the Royal Society press office to national UK, international, and scientific news outlets to promote your work.

Author's Response to Decision Letter for (RSOS-201086.R0)

See Appendix B.

Decision letter (RSOS-201086.R1)

Dear Mr Alexiadis,

It is a pleasure to accept your manuscript entitled "Transformation of measurement uncertainties into low-dimensional feature vector space" in its current form for publication in Royal Society Open Science. The comments of the reviewer(s) who reviewed your manuscript are included at the foot of this letter.

You can expect to receive a proof of your article in the near future. Please contact the editorial office (openscience@royalsociety.org) and the production office (openscience_proofs@royalsociety.org) to let us know if you are likely to be away from e-mail contact – if you are going to be away, please nominate a co-author (if available) to manage the proofing process, and ensure they are copied into your email to the journal.

on behalf of Professor Len Thomas (Associate Editor) and R. Kerry Rowe (Subject Editor)
openscience@royalsociety.org

Appendix A

Response to editor's comments on

Transformation of measurement uncertainties into low-dimensional feature vector space

A. Alexiadis, S. Ferson, E. A. Patterson

Comment from editor:

Thank-you for your submission. While the method you propose may be sound, I simply do not understand the reason why one may want to use it, compared to the much more simple alternative of a parametric bootstrap. As I understand your motivation, the goal is to transform measurement error, either in the form of known distributions or a set of bounds (measured value $\pm 2 \mu_{meas}$) coupled with a uniform distribution, into a characterization of uncertainty on a reduced feature space that is created by a deterministic set of operations. A straightforward way to achieve this is a parametric bootstrap - i.e., to sample multiple times from the measurement error distributions and for each sample repeat the deterministic operations to produce multiple realizations of the reduced feature space; the distribution of these realizations then forms a simulation-based approximation of the distribution of uncertainty in the output given the input measurement error distribution.

Authors' response:

We thank the editor for their comments on our manuscript which have allowed us to focus more sharply our introduction and explain our motivation. The editor proposed that a parametric bootstrap would be a straightforward way to achieve our goal. This is based on the editor's understanding of our goal as '...to transform measurement error, either in the form of known distributions or a set of bounds coupled with a uniform distribution, into a characterization of uncertainty on a reduced feature space...' However, we have made no assumptions about the probability distribution of the measurement error – it is our unknown likelihood in the approximate Bayesian computation – and our intention was to create a distribution-free method that works without such an assumption. Thus, we believe that a misunderstanding has arisen about our goal because the underlined statement above does not apply and without this statement, it is not possible to construct a parametric model that can be randomly sampled in a bootstrap.

Of course, if a misunderstanding has occurred then it must be our fault for failing to explain our goal and motivation clearly. Hence, we have revised the abstract, introduction, proposed methodology and conclusions in an attempt to clarify the situation and highlighted the changes in the revised manuscript.

We would also like to apologise for our delay in responding to the editor's comments which have been caused by our corresponding author focussing on submitting his PhD thesis on schedule under lockdown conditions in the UK. We hope that the editor will permit our revised manuscript to enter the review process.

Appendix B

Transformation of measurement uncertainties into low-dimensional feature vector space [ID RSOS-201086]

Responses to Comments from the Editor:

Please accept my apologies for the longer-than-usual time to provide a response on this manuscript. It has proven very difficult to obtain reviewers. We have received one review; the reviewer was broadly supportive of publication and I concur. Based on my reading of their comments I am recommending acceptance with minor revisions. In your resubmission, please provide a point-by-point response to their comments, and to mine below. I will review the changes and your responses and make a final decision – I do not think the manuscript will need to go for review again.

- Thank you for your perseverance in obtaining reviews. We have addressed each comment below and highlighted the changes in the manuscript.

Regarding the reviewer's last comment "In case a measurement uncertainty is uniform ... the error distribution is skewed over the field" I'm afraid I cannot understand the comment. I hope you can, but in case you cannot either, just let me know in your response letter.

- We believe we have understood the point being made and have tried to address it below.

Additional points to respond to are:

1. Please ensure you provide all data and code required to reproduce the results in this paper, in line with the journal's policy. Failure to do this will result in rejection of the paper.

- We have added the URL for the location of our code and data.

2. Page 11 "Similarly, with knowledge of the probability distribution of the measurement error, it might be possible to use a parametric bootstrap to sample the distribution..." Thank-you for mentioning this. If the sample size of measurements is sufficient, might it also be possible to use a non-parametric bootstrap, sampling the measurements with replacement, to quantify uncertainty?

- Possibly. However, we believe that applying non-parametric bootstrap with geostatistics-based techniques will be prohibitive when thousands of point measurements are available due to the difficulty associated with establishing the covariance matrix. We have added a sentence to this effect in the manuscript.

3. Page 16, last sentence "... and the significance of differences between data fields..." What significance do you mean, exactly, here? Presumably not statistical significance in the classical hypothesis testing meaning. Assuming you mean "importance" can you please give an example?

- Thank you for highlighting our ambiguous use of English. Yes, we meant 'importance'. We have changed the manuscript and given an example as requested.

Response to Comments from Reviewer:

- We would like to thank the reviewer for their comments. We have responded below and accordingly revised the manuscript highlighting our changes.

The paper proposed an interesting method using approximate Bayesian computation, while the basic concept/idea of reducing the dimensionality of data matrices to feature vectors was published before. The paper mentioned "decision-making" a number of times, but it is not clear that what decision-making it can help, to what degree/confidence level? the reviewer recommend to provide an example to elaborate further on a real decision-making case, how can this concept/method help to determine the consequence/risk of a decision-making using the low dimensional feature vector space?

- We have amended the manuscript to amplify the contribution to decision-making that we believe our method could make and also added examples in the context of the case studies presented in the paper.

In case a measurement uncertainty is uniform over the field of measurements, for example depending on stress gradient or surface finish, the error distribution is very skewed over the field, can the proposed concept and method still determine a reasonable error distribution?

- Our use of equation 2 assumes that the measurement error is Gaussian; so, if the measurement error at each pixel location were very skewed then equation 2 would need to be modified based on knowledge of the form of the measurement error. We have added a comment in the manuscript to highlight this limitation in our technique.